# Carbohydrate-Containing Low Molecular Weight Metabolites of Microalgae

**DOI:** 10.3390/md21080427

**Published:** 2023-07-28

**Authors:** Valentin A. Stonik, Inna V. Stonik

**Affiliations:** 1G.B. Elyakov Pacific Institute of Bioorganic Chemistry, Far Eastern Branch of the Russian Academy of Sciences, Pr. 100-letya Vladivostoka 159, 690022 Vladivostok, Russia; stonik@piboc.dvo.ru; 2A.V. Zhurmunsky National Scientific Center of Marine Biology, Far Eastern Branch, Russian Academy of Sciences, ul. Palchevskogo 17, 690041 Vladivostok, Russia

**Keywords:** microalgae, glycoconjugates, arsenicals, galactolipids, steryl glycosides, prymnesins

## Abstract

Microalgae are abundant components of the biosphere rich in low molecular weight carbohydrate-containing natural products (glycoconjugates). Glycoconjugates take part in the processes of photosynthesis, provide producers with important biological molecules, influence other organisms and are known by their biological activities. Some of them, for example, glycosylated toxins and arsenicals, are detrimental and can be transferred via food chains into higher organisms, including humans. So far, the studies on a series of particular groups of microalgal glycoconjugates were not comprehensively discussed in special reviews. In this review, a special focus is given to glycoconjugates’ isolation, structure determination, properties and approaches to search for new bioactive metabolites. Analysis of literature data concerning structures, functions and biological activities of ribosylated arsenicals, galactosylated and sulfoquinovosylated lipids, phosphoglycolipids, glycoside derivatives of toxins, and other groups of glycoconjugates was carried out and discussed. Recent studies were fundamental in the discovery of a great variety of new carbohydrate-containing metabolites and their biological activities in defining the role of microalgal viral infections in regulating microalgal blooms as well as in the detection of glycoconjugates with potent immunomodulatory properties. Those discoveries support growing interest in these molecules.

## 1. Introduction

The low molecular weight natural products such as glycerol- and sphingoglycolipids, glycosylated fatty alcohols derivatives, carbohydrate-containing toxins, steroid and other glycosides, and some ribose-containing arsenicals belong to the group of molecules called glycoconjugates. Being key cellular components and participating in both inter- and intracellular communications, metabolites of this type are ubiquitous in nature and influence important biologic properties such as fluidity and permeability of biomembranes, stimulation of apoptosis, and defense against predators. Microalgae are one of the primary producers of glycoconjugates, which along with nutritionally important polyunsaturated fatty acids of ω-3 (n-3) series, pigments, antioxidants, and other bioregulators, are accumulated via food chains in many marine organisms, including edible species of fish and mollusks. Glycoconjugates are a portion of many various chemicals produced by microalgae.

Microalgae represent widespread eukaryotic microorganisms that populate fresh, brackish sea waters and bottom sediments and have economic significance that grows year to year [1]. A part of microalgal species are symbionts living in host organisms, for example, in corals. Microalgae are photosynthesizing organisms that provide about half of the atmospheric oxygen and the main part of organic substances on our planet and support more than 60% of the total primary production in marine ecosystems [2]. Diatoms, dinoflagellates, coccolithophores, microscopic green and red microalgae, and representatives of some smaller taxa are related to different groups of microalgae, in total belonging to more than 50,000 species that provide a significant biochemical diversity. Glycoconjugates are also produced by cyanobacteria (cyanophytes), which, like algae, are autotrophs and are capable of photosynthesizing and releasing oxygen but have more similarity with bacteria than with low plants. Therefore, we considered not covering this group of organisms in our review.

Glycoconjugates in microalgae form complex and difficult-to-separate mixtures of biomolecules, which are increasingly studied by the HPLC/MS regardless of the method’s inability to fully characterize molecule stereochemistry. That is why, in this review, we give structural information only for such carbohydrate-containing metabolites that have been obtained and purified in amounts sufficient to carry out structure analyses using NMR methods and chemical transformations. In other cases, the structures of the studied molecules are described by their abbreviations. 

Quite recently, we published review articles concerning low molecular weight metabolites from diatoms [3] and studies on microalgal glycoconjugates such as steryl glycosides and sphingosines [4]. The current work covering a period from 1989 to 2022 is the result of an expansion and continuation of our preliminary efforts. 

## 2. Glycosylated Arsenicals

After the discovery of arsenic-containing compounds in fishes in the 1920s [5], a variety of close related natural products of this type were found in macro- and microphytes as well as in different marine animals. Arsenic specification, toxicity and metabolism of these compounds in microalgae were discussed in the review by Wang et al. [6]. Typically, arsenic is present in seawater in a concentration of 1–2 μg per liter and is incorporated from the environment into living systems as a result of metabolic processes, leading to more bioavailable products. A variety of water-soluble arsenic species in marine microalgae includes not only arsenosugars but also inorganic species such as arsenate (V) and its methylated derivatives. Chlorophytes produce glycerol- and phosphate arsenosugars, whereas glycerol-, phosphate-, and sulfate-containing arsenoribosides, as well as dimethyl arsenates, are more common in heterokontophytes, for example, diatoms. Some arsenic-containing metabolites such as glycerol arsenosugars (**1**), phosphate arsenosugars (**2**), sulfate arsenosugars (**3**), and sulfonate arsenosugars (**4**) (Figure 1) along with glycosylated lipid-soluble arsenolipids from microalgae belong to glycoconjugates and together make up a part of a relatively large class of natural compounds, so-called arsenicals.

In this class, more than 70 arsenosugar compounds with established structures are known. Although their inorganic moieties can include either As (III) or As (V), the latter forms predominate. Contrary to arsenosugars, arsenolipids rarely contain monosaccharide residues and represent lipid-soluble and frequently volatile natural products. Both arsenosugars and arsenolipids can be transferred via the food chain into invertebrates and higher animals, including humans. These compounds are more or less detrimental due to their toxicity and cytotoxicity. 

Arsenic toxicity is a global problem. Millions of people are exposed to As-containing substances through drinking water and food. High arsenic content can induce chronic toxicity and, in some cases, cancer. In some individuals, acute, subacute, and chronic poisonings, characterized by skin lesions, cardiovascular dysfunction, and multi-organ failure, can be developed [7]. 

Historically, As-containing preparations such as Salvarsan (arsphenamine) for the treatment of syphilis and arsenic trioxide for cancer were among the drugs that found widespread application in medicine. The useful properties of arsenic trioxide were recently rediscovered [8,9].

Arsenosugars (**1**–**4**) were found in different diatom and green microalgae such as *Phaeodactylum tricornutum*, *Thalassiosira pseudonana*, *Ostreococcus tauri*, *Dunaliella tertiolecta*, and others [10,11,12,13]. Generally, arsenosugars contain ribose or its derivatives, and marine microorganisms produce arsenoribosides in higher concentrations in comparison to fresh-water microorganisms. Total arsenic concentrations in algae (0.1–10 µg/g of dry mass) is 10–100 times higher than in terrestrial plants [14].

Lipid-soluble fractions of arsenic-containing compounds are characterized by a great diversity of metabolites. Frequently, arsenicals are analogous to common phospholipids with the change of phosphorus for arsenic; more than 100 different lipid-soluble arsenicals were isolated [15]. Very recently, Chinese scientists reported that over 300 species of naturally occurring organoarsenicals were identified by modern analytical methods, particularly HPLC/MS, as a promising option [16].

The unicellular green alga *Dunaliella tertiolecta* is one of the most known and capable of photosynthesizing marine microorganisms belonging to the order Chlamydomonadales. This species, like other lower plants of the genus *Dunaliella*, survives in hypersaline environments. Along with several previously known arsenosugars, the novel lipid-soluble arsenical phytyl 5-dimethylarsinoyl-2-O-methyl-ribofuranoside (AsSugPytol546) (**5**) (Figure 2) was detected by high-resolution electrospray mass spectrometry (HR ESI MS) in cultured microalgae *D. tertiolecta* as well as in extracts from oceanic phytoplankton. 

In order to obtain compound **5** in quantities sufficient for use in NMR spectroscopy and to determine its structure, this microalga was cultured in an arsenate-enriched medium. About 2 g of cultured dried cells were used for solvent partitioning, column chromatography on silica gel and preparative reverse-phase HPLC. As a result, it was possible to obtain only a trace amount of the target compound (about 100 micrograms). Nevertheless, subsequent NMR analysis and comparison of the tandem mass spectrum (MS/MS) of the obtained arsenolipid with those of the synthesized model methyl-5-dimethylarsinoyl-2-O-methylriboside allowed the establishment of the structure of **5** and showed that it contains phytol as an aglycone. The Glycon moiety of this arsenolipid consisted of 5-dimethylarsinoyl 2-O-methylribose, and, therefore, a methoxy group replaces a sugar hydroxyl in this unusual monosaccharide. This structural feature, characteristic of some RNAs, was not previously found outside the RNAs world [17].

A possible pathway of the biosynthesis of AsSugPhytol546 was proposed to include the introduction of an arsenic group into the ribosyl moiety of S-adenosylmethionine (**6**→**7**) followed by 2-O-methylation by the corresponding methylase to yield **8**. The loss of an adenosyl residue at the catalysis by adenosyl nucleosidase to give the ribose derivative (**9**), and finally, the interaction of the latter with phytyl diphosphate, catalyzed by riboside phytolsynthase, led to **5** (Figure 2) [18]. 

**Figure 2 marinedrugs-21-00427-f002:**
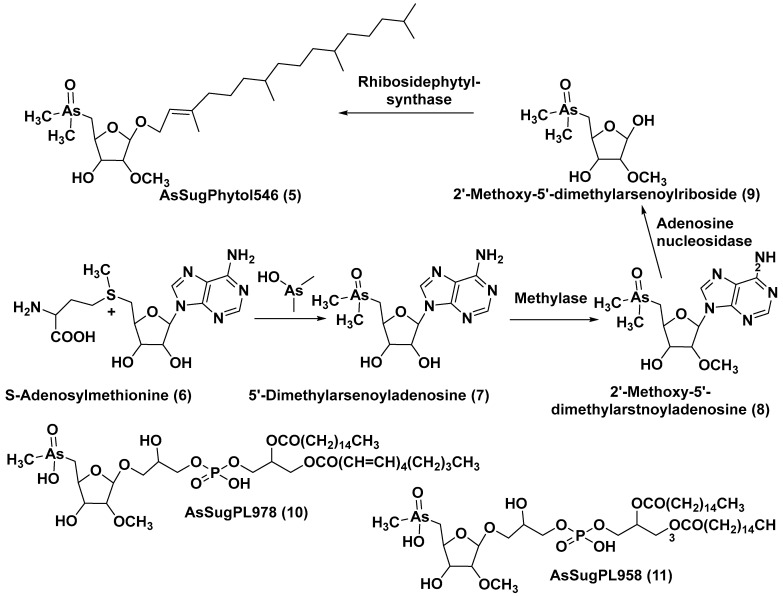
Biosynthesis of **5** and structures of other arsenolipids found in *D. tertiolecta* (**6**–**11**) [18,19].

In addition to the major arsenolipid **5**, a new metabolite, designated as AsSugPL978 (**10**), and related AsSugPL958 (**11**) were identified in the same culture. It is of particular interest that more than half of the total arsenic is present in *D. tertiolecta* as lipid-soluble forms. Although the relative amount of total arsenolipids remained constant in different cultivation conditions, an increase in the content of hydrophilic arsenosugars was discovered under various As/P regimes when As concentrations were increased in the medium [19].

The accumulation of arsenic species from the environment into the above-mentioned microalgae stimulates the biosynthesis of various arsenic-containing glycoconjugates. Some arsenic-containing natural products of this type can be found in many popular sea foods. Further progress in the search for this type of compound is connected with the use of HPLC/MS after chemical derivatization [19]. It should be noted that some oleaginous microalgae, such as *Scenedesmus* sp. IITRIND2 are super accumulators of arsenic. Being stressed by arsenic intake, arsenic-tolerant species modulate their own cellular processes, enhance lipid production and accumulate large lipid droplets. The biodiesel obtained from such microalgae was comparable to plant oil methyl esters and had a high cetane number and oxidative stability [20].

## 3. Glycolipids

The evolution of organisms and their biochemical systems, and in particular processes of photosynthesis from cyanobacteria to microalgae, attracts attention to autotrophic microalgae and their metabolites. Glycolipids, as products of mixed biogenesis, containing both lipid and carbohydrate moieties, represent microalgae metabolites of particular interest and importance.

### 3.1. Galactolipids of Microalgae

Galactolipids, as a subgroup of glycolipids, are major membrane constituents of microalgal plastids, which play an important role in harvesting and converting sunlight energy. The presence of galactolipids in thylakoids of plastids supports maximal photosynthetic efficiency, as was shown by the indication of specific interactions between the galactolipid head groups and photosynthetic protein complexes in plants. Moreover, it explains the preference of galactose in thylakoid lipids not only in lower but also in many higher plants [21]. Due to their high content in thylakoid membranes and the wide distribution in lower and higher plants, galactolipids are considered to be the most abundant lipid class in the biosphere [22].

Membranes of plastids are mainly comprised of monogalactosyldiacylglycerols (MGDG, **12**) and digalactosyldiacylglycerols (DGDG, **13**), along with less abundant sulfoquinovosyldiacylglycerols (SQDG, **14**). Galactosylmonoacylglycerols (MGMG) **15** and **16** differ from each other in the position of an acyl group attached to a glycerol residue (Figure 3). 

Such metabolites were found in microalgae much less frequently than MGDG and DGDG. Structurally, wide-distributed in plants MGDGs are characterized by the presence of one D-galactose, β-anomerically linked to the sn-3 position of the glycerol backbone. DGDGs contain an additional terminal D-α-galactose attached to the inner β-galactose residue by a 1,3-bond. SQDGs consist of α-D-6-sulfoquinovose and 1,2-diacylglycerol linked to each other in the same manner as in MGDG.

Metabolites of general formulae **12** and **13** maintain chloroplast morphology and survival of plants under abiotic stresses. Lipids, making up the plastid membranes in algae, are biosynthesized by either prokaryotic or eukaryotic pathways located within the plastids and in the endoplasmic reticulum, respectively. Thylakoid glycolipidome was formed in the process of evolution from cyanobacteria via lower plants to higher plants. In microalgae, glycolipids are enriched by polyunsaturated fatty acids (FAs) belonging to the ω-3 series. Fatty acid compositions of glycolipids differ from species to species and depend on conditions of microalgae growth. It is known that glycolipids of microalgae and their metabolites have an important nutritional significance and often possess cytotoxic and anti-inflammatory properties [23].

It was shown that galactolipids of dinoflagellates are diverse in structure, and this diversity depends on the origin of chloroplasts in these microalgae. A probable general scenario of the evolution of galactolipids in chloroplasts of algae and other plants was discussed by Sato and Awai [24]. They suggested that chloroplasts of plants, including algae, share a common origin with cyanobacteria. The Biosynthesis of MGDG is realized with the participation of glycosyltransferases, encoded by genes MGD-1,2,3. They catalyze the transfer of the α-galactosyl residue from uridine diphospho-galactose (UDP-Gal) to diacylglycerol (DAG). Conversion of MGDG into DGDG is catalyzed by epimerases, encoded by genes DGD-1,2. One more type of enzymes, desaturases, catalyze the introduction of additional double bonds in acyl substituents of DAG. In contrast with cyanobacteria, the biosynthesis of MGDG from diacylglycerols in chloroplasts of plants takes place without the formation of intermediate glucosyldiacylglycerols [24].

The evolution of photosynthesizing complexes included two branches: one from primary endosymbiosis in symbiotic cyanobacterium that have primary chloroplast, and the second one that includes symbiotic unicellular algal eukaryote with a secondary plastid [25]. Carbohydrate-containing metabolites of MGDG and DGDG groups can also be exported into other subcellular compartments. 

Structural determination of molecular forms of these lipids must include the indication of fatty acid substituent positions attached to glycerol, the determination of stereochemical peculiarities such as configurations of asymmetric centers in glycon and aglycon moieties and the nature of the acyl substituents. The procedure is performed after the separation of glycolipid mixtures, isolation and purification of a sufficient amount of target substances, followed by NMR analyses supported by mass-spectroscopy and chemical transformations.

Italian scientists applied a simpler, based on ^1^H NMR spectroscopy, approach to identify and quantify glycolipid components of three microalgal species (*Thalassiosira weissflogii*, *Cyclotella cryptica* and *Nannochloropsis salina*). MGDG, SQDG, and DGDG lipids were found in each of the studied species. For structure analysis, signal areas and chemical shifts in spectra characteristic of various fragments of these algal glycolipids were compared with calibrated proton NMR signals of an external standard [26]. This procedure, known as the ERETIC method [27], is applicable to glycolipid mixtures without their preliminary separation and may be used for the characterization of glycolipids. 

One of the first cases of studies on macroalgal galactolipids aimed at the determination of complete structures of their molecular forms was described by Yasumoto et al. [28], who isolated the 1-O-6,9,12,15-octadecatetraenoyl-3-O-[β-D-galactopyranosyl-

-(1→6)-O-β-D-galactopyranosyl]-sn-glycerol (**17**) along with an impurity of 1-O-3,6,9,12,15-octadecapentaenoyl-3-O-[β-D-galactopyranosyl-(1→6)-O-β-D-galactopy-ranosyl]-sn-glycerol from the microalga *Prymnesium parvum* as hemolytically active substance (so-called Hemolysin I) (Figure 4). They used chromatographic separation of galactolipid fraction, enzymatic hydrolysis of obtained compounds and NMR studies to solve the problem of the structure determination of **17** belonging to the group of MGMGs [28].

Kitagawa’s research group found α-galactose in galactolipids in microalga *Heterosigma akashiwo* at the beginning of the eighties [29]. This microalga, named “akashiwo” from the Japanese word “red tide”, episodically causes the events of blooms in Japanese inner sea waters. Galactolipids from *H. akashiwo* were shown to consist of several molecular forms, all bearing polyunsaturated fatty acids of ω-3 (n-3) series as acyl substituents. Cultured dinoflagellates were sonicated and extracted. The extract was subjected to silica gel column chromatography to obtain MGDG, DGDG, and SQDG fractions. Further separation of two first fractions using HPLC on a Zorbax ODS column in methanol-water mixtures gave *Heterosigma*-glycolipids I-IV (**18**–**21**) (Figure 4). The treatment with sodium methylate to obtain FA methyl esters allowed determining FA compositions of these glycolipids. At this stage, the known glycerol galactoside was also isolated. All obtained compounds were studied by NMR and MS methods and shown to contain β-D- and α-D-galactose residues (compounds **18**–**20** and **21**, respectively) in carbohydrate moieties.

To establish the positions of two different FA residues in **18**, enzymatic hydrolysis, using lipase type XIII from *Pseudomonas* sp., was carried out. As a result, a monoacyl derivative (**22**) with a single acyl substituent at position 2 was obtained. Compound **20** gave methyl 6,9,12,15-octadecatetraenoate after the treatment with NaOMe-MeOH. Thus, it became clear that the corresponding FA was located at position 2 and, therefore, the total structure of **16**. Similarly, the same product (**22**) was also obtained from *Heterosigma*-glycolipid II (**19**). Positions of acyl substituents in *Heterosigma*-glycolipid IV (**21**) were established as a result of its transformation first into monoacyl derivative (**23**) by enzymatic hydrolysis, with subsequent removal of 5,8,11,14,17-eicosapentaenoate by sodium methylate in MeOH as methyl derivative [29].

Glycolipids are known to be part of plastids as well as cytosolic lipids. Taking into account the origin of plastids in microalgae, it is suggested that structures of galactolipids be connected with primary, secondary or tertiary symbioses. Microalgal species originating from different symbiotic pathways are characterized by differences in FA compositions.

Dinoflagellate *Heterocapsa circularisquama*, the alga which caused mortalities of pearl oysters as well as other mollusks in Japanese waters, and several other microalgae were studied for structures and activities of their galactolipids (**24**–**32**) [30,31,32,33,34,35]. The obtained data are summarized in Table 1. 

Thus, microalgal galactolipids are characterized by a chemical diversity associated mainly with acyl substituents and their positions in the glycerol moiety and less often in carbohydrate fragments as well as with different carbohydrate moieties. Polyunsaturated FAs of ω-3 series are the most characteristic constituents of these galactolipids. A great structural diversity determines a set of biological effects of galactolipids from microalgae. However, in the majority of cases, galactolipids were studied without separation of their fractions, for example, hemolytic fractions from two species of dinoflagellates belonging to the *Gymnodinium* genus [36]. 

Most common polyunsaturated FAs in glycolipids from microalgae, particularly dinoflagellates, were identified as C18 compounds, but longer-chain fatty acids were also found in the corresponding metabolites [37,38]. Leblond and Chapman [39] developed a convenient procedure for the isolation of galactolipids from lipid mixtures by column chromatography with a gradual increase in the polarity of eluting systems. They obtained galactolipids from lipid extracts by the elution from a UniSil silica column with acetone as eluent. The careful analysis allowed for examination of the distribution of C28 and C18 fatty acids in dinoflagellates, and it was shown that 28:7(n-3) and 28:8(n-3) FAs are located in phospholipids, while 18:5(n-3) and 18:4(n-3) are the constituents of galactolipids of chloroplasts. Long-chain fatty acids (C28) were probably biosynthesized in the cytoplasm, while C18 FA galactolipids were within chloroplasts [39]. For a long time, the 18:5(n-3) fatty acid was considered a chemotaxonomic marker of Dinophyta, but later this FA was found in other microalgal taxa such as Haptophyceae, Eustigmaphyceae [40,41], and Raphidophyceae [42,43].

Structure analysis of galactolipids was significantly facilitated by the wide application of HPLC, particularly HPLC/MS methods [44,45], as well as by the development of the latter approach into HPLC/ESI MS/MS analysis [46]. This technique was repeatedly used and allowed the rapid identification of related structures in different biological objects. In 2003, Guella et al. [47] developed and used this approach to establish positions of acyl substituents in galactolipids. Collision-induced ionization dissociation of the components of MGDG and DGDG mixtures, previously separated by high-performance HPLC, was performed by tandem positive-mode ESI MS/MS and compared with ESI MS/MS of sn-2 lysoglyceroglycolipids, which were obtained by regiospecific enzymatic hydrolysis of corresponding diacylglycerols using *Rhizopus arrhizus* lipase. For asymmetrically disubstituted galactolipids such as (**33**), the positive mode MS2 of [M + H]^+^ ions (**34**) indicated the preferred loss of acyl substituent from the sn-1 position (ion 36) as illustrated below (Figure 5).

The authors have formulated the following rule for the determination of positions of acyl substituents in galactolipids: “the positional distribution of the acyl chains in galactolipids can be established knowing that, in positive-ion mode ESI MS2 measurements, the loss of the carboxylic acid linked to the sn-1 glycerol position (leading to ion **35**) always produces a more intense peak than that derived from the loss of the sn-2 linked acyl chain” (ion **36**). This rule is applicable to the both MGDGs and DGDGs [48].

Many dinoflagellates contain the carotenoid pigment peridinin, the principal constituent of water-soluble light-harvesting peridinin-chlorophyll-protein complex in chloroplasts. Peridinin was first isolated over 100 years ago. Application of ESI MS to analyze the MGDGs and DGDGs of 35 peridinin-containing species from the class Dinophyceae and particularly the use of tandem mass-spectrometry allowed the determination of the positional distribution of FAs associated with different classes of galactolipids. The examined dinoflagellates were divided into two clusters based on the molecular forms of MGDGs and DGDGs presented. The first cluster possessed 18:5/18:4 MGDG (sn-1/sn-2), 18:5/18:5 MGDG, 18:4/18:4 DGDG, and 18:5/18:4 DGDG as major forms, while the microalgae belonging to the second cluster had 20:5/18:4 and 20:5/18:5 MGDG, 20:5/18:4 and 20:5/18:5 DGDG as major forms. The majority of peridinin-containing dinoflagellates contain secondary plastids, presumably of red algal origin [49]. 

No other microalgae possess so much diversity of their galactolipids as dinoflagellates. During their evolution, some dinoflagellates lost ancestral peridinin-containing plastids several times but then gained new endosymbionts and restored the capability to photosynthesize as a part of tertiary endosymbiotic events [50,51]. Thus, microalgae adapted the biochemical pathways retained from the ancestral plastid for transcript processing in their current plastids. It was suggested that, as a result of the event, the genera *Karenia* and *Karlodinium* possess plastids of haptophyte origin; *Lepidodinium* (formerly *Gymnodinium chlorophorum*) possess plastids of green algal, possibly prasinophyte origin; *Kryptoperidinium* (formerly *Peridinium foliaceum*) has endosymbiont plastids of pennate diatom origin. Glycoconjugates of these species with aberrant plastids were studied by ESI MS and ESI MS/MS. *L. chlorophorum* and *K. brevis* contained 18:5/18:5 MGDG (like it was observed in many other peridinin-containing dinoflagellates), along with other forms of MGDG and DGDG, previously not found in these microalgae. *L. chlorophorum* was found to possess 18:5/16:4 MGDG and 20:5/16:4 DGDG. The 18:5/14:0 MGDG and DGDG molecular forms were found in *K. brevis*. For comparison, the corresponding compounds from green microalgae *Tetraselmis* sp., the haptophyte *Emiliania huxleyi,* and the diatom *Navicula perminuta*, which are thought to have common ancestors with those of aberrant dinoflagellates, were also examined by the same methods. It was shown that MGDG and DGDG compositions of the *K. foliaceum*/*N. perminuta* pair were almost the same, whereas, in the *L. chlorophorum*/*Tetraselmis* sp. and *K. brevis*/*E. huxleyi* pairs, the MGDG and DGDG compositions were similar, but not in all galactolipid components matched. In general, these data confirmed the hypothesis regarding the evolution of plastids and plastid glycolipids in some microalgae as a consequence of the tertiary symbiosis phenomenon with the participation of haptophyte, green, and diatom microalgae, respectively [51]. 

A recent analysis of galactolipid compositions of the microalgae belonging to the genus *Amphidinium*, one of the largest genera of Dinophyta, showed that its metabolites could be basal to those of a group of peridinin-containing dinoflagellates. This hypothesis was proposed by Leblond et al. [52].

It is of particular interest that symbiont dinoflagellates belonging to the genus *Symbiodinium* (so-called zooxanthellae) are the largest class of obligatory endosymbionts in marine invertebrates, first of all, in cnidarians such as corals, sea anemones, and jellyfish. Dinoflagellates *Symbiodinium* spp. were also found in some species of sponges, flatworms, mollusks, foraminifera and ciliates. The symbiosis of corals with zooxanthellae is an example of mutualistic symbiosis in that both partners benefit. It is known that corals having zooxanthellae calcified and increased the photosynthetic fixation of CO_2_ much faster than those without dinoflagellates. Awai et al. [53] determined characteristic features of microalgal MGDG, DGDG and SQDG in two *Symbiodinium* strains isolated from the jellyfish *Cassiopea ornata* and the giant clam *Tridacna crocea*. It was shown that there is a transport of FAs from hosts to symbionts that affects total lipid balance in symbiont dinoflagellates [54]. Thirteen molecular species of MDGDs and ten such species of DGDGs in zooxanthellae from the tropical soft coral *Capnella* sp. were studied by high-resolution tandem mass spectrometry. As a result, unique molecular species of MGDGs (16:4/18:5) and 18:4/18:4, 18:4/20:5, and 16:2/22:6 molecular species of DGDGs were indicated [55]. The main FAs in the zooxanthellae symbiotic of a soft coral *Sinularia* sp. were identified as 18:4n-3, 20:5n-3, and 22:6n-3. Polar lipids with 18:4n-3, 18:5n-3, and 20:5n-3 FAs are considered the most characteristic of these symbionts [56]. 

Chloroplastic MGDG and DGDG were studied by the positive-ion ESI MS and ESI MS/MS in four peridinin-containing, cold-adapted dinoflagellates (*Gymnodinium* sp., *Peridinium aciculiferum*, *Scrippsiella hangoei*, and *Woloszynskia halophila*), grown at 4 °C. Dominant forms contained only C18 fatty acids with the exception of the *Gymnodinium* sp. from the Baltic Sea that contained a 20:5/18:5 form of DGDG. Each cold-adapted dinoflagellate contained both 18:5/18:5 and 18:5/18:4 DGDG, while the majority of warm-adapted dinoflagellates contained only 18:5/18:4 DGDG. The presence of the 18:1/14:0 trigalactosyldiacylglycerol (TGDG) was also established as the dominant galactolipid in *Gymnodinium* sp. Probably, this metabolite contains an additional D-galactopyranosyl unit, linked by α-(1→6) bond to the terminal galactose of DGDG [57].

Diatoms (the phylum Bacillariophyta), along with dinoflagellates, are the largest taxa of primary producers in the oceans. Structural studies on galactolipids from diatoms were initiated in 1993 [58]. The diatom *Phaeodactylum tricornutum*, which can exist in different morphotypes and change cell shape, predominately has the same classes of galactolipids (MGDG, DGDG and, in smaller amounts, SQDG) as other microalgae. This species contains EPA (20:5n-3) at the sn-1 position and C16:1, C16:3, or C16:4 FAs at the sn-2 position in these lipids [59]. 

Similar fatty-acid distribution was established in many other diatom species. FA composition of diatoms was studied in many papers, and the majority of scientists noted the high content of galactolipids in total lipid fractions. For example, Dunstan et al. [60] examined FA compositions of 14 diatom species from the genera *Skeletonema*, *Thalassiosira*, *Navicula*, and *Haslea*. In general, the diatoms possessed larger amounts of C14:0, C16:0, C16:1, and C20:5 fatty acids when compared with dinoflagellates. The major fatty acids in most species were identified as 14:0, 16:0, 16:1(n-7) and 20:5(n-3). Zhukova and Aizdaicher [61] have concluded that the marker of Bacillariophyceae is the prevalence of 16:1(n-7) over 16:0, high levels of 14:0, 20:5(n-3) FAs, at that C16 unsaturated FAs contain double bonds at (n-4) and (n-1) positions. 

Using ultra-performance liquid chromatography-electrospray ionization-quadrupole-time of flight-mass spectrometry, Yan et al. [62] examined MGDGs and DGDGs in three strains of the diatom *Skeletonema* sp. The predominant species of MGDGs were identified as those containing 16:3, 20:5, 16:1, and 16:3 FAs. Three main DGDGs exist in this microalga as 20:5/16:1(sn-1/sn-2), 20:5/16:2, and 16:1/16:1 molecular forms. Based on the identification of FA residues in the sn-2 position, it was proposed that MGDGs and DGDGs are biosynthesized within chloroplasts by prokaryotic pathways exclusively. 

Most diatoms produce a set of photosynthetic pigments, including green chlorophylls along with yellow or brown carotenoids, that provides them a green or golden-brown color. The ‘blue’ pennate diatom *Haslea ostrearia* synthesizes and releases into the environment the water-soluble polyphenolic, non-photosynthetic pigment marennine. In their studies on centric and pennate diatoms, *H. ostrearia*, *P. tricornutum*, *Skeletonema marinoi*, *Navicula perminuta*, and *Thalassiosira weissflogi,* Dodson et al. [63] focused on the marennine-producing pennate diatom *H. ostrearia*. Application of ESI MS for analysis of their MGDG and DGDG structures indicated this microalga contains primarily C18/C16 or C18/C18 forms of MGDG and DGDG in contrast to *S. marinoi*, *T. weissflogii*, and *P. tricornutum*, that have C20/C16 and C18/C16 molecular forms of these galactolipids.

Raphidophyte algae (Raphidophyceae) include brown- and green-pigmented taxa. Compositions and positional distribution of FAs in MGDG and DGDG were examined using ESI MS and ESI MS/MS in the positive ions mode. Brown-pigmented strains of the genera *Chattonella*, *Fibrocapsa*, and *Heterosigma* primarily possessed 20:5/18:4 MGDG and 20:5/18:4 DGDG, while the green-pigmented *Gonyostomum semen* had these forms along with 18:3/18:4 MGDG and DGDG which are characteristic of green algae [64].

The microscopic red alga *Cyanidioschyzon merolae* from an Italian hot spring has an extremely simple FA composition with only C16:0, C18:0, and C18:1 n-9 as major FAs in their galactolipids. Interestingly, the survival of microalgae in extreme conditions of a hot spring was associated with the loss of desaturase activity in plastids and, as a consequence, with the presence of mainly saturated FAs in galactolipids [65].

Thus, galactolipids are diverse in both carbohydrate fragments and FA compositions. In many cases, these carbohydrate-containing metabolites contain residues of polyunsaturated FAs, often attached to the sn-1 position in glycerol moiety. Being mandatory components of the photosynthetic apparatus, galactolipids serve as a reservoir of essential polyunsaturated fatty acids necessary for human health and exhibit diverse biological activities that open up prospects for their obtaining from cultured microalgae to use these compounds in medicine.

### 3.2. Sulfoquinovosyl-Containing Glycolipids

Along with galactolipids, sulfur-containing sulfoquinovosyldiacylglycerols (SQDGs) are also known as essential metabolites that are localized in thylakoid membranes of plastids in many organisms capable of photosynthesis, including microalgae. These metabolites are biosynthesized from uridine diphosphate glucose in two stages: one) by the formation of uridine diphosphate sulfoquinovose, catalyzed by UDP-sulfoquinovose synthase; two) the subsequent reaction with diacylglycerols, catalyzed by the glycosyltransferase SQDD-synthases [66]. The chemical diversity of SQDGs is determined mainly by FAs linked to glycerol. A distinctive feature of SQDGs consists of a polar sulfonic acid residue bonded with C-6 in a quinovose unit, which provides the high polarity of these compounds.

Regiochemical assignment in sufoquinovosyl diacylglycerols, like that in other galactolipids, can be established using tandem mass-spectrometry but in negative mode with collision-induced dissociation ionization. Ion peaks corresponding to the loss of fatty acid from the sn-1 position of the studied sulfoglycolipids in these spectra are more intense in comparison with those resulting from the loss of neutral FA from the sn-2 position [67].

Blooms of the chloromonad *Heterosigma carterae* (formerly *H. akashiwo*) (Raphidophyceae) were observed in different areas of the World Ocean. The isolate 102R of this microalga was cultured in natural seawater, and four SQDGs (**37**–**40**) from the extract of the obtained culture were partially purified and structurally identified using LC/MS/MS technique and NMR analysis. SQDG fractions were treated with BF_3_ in methanol with liberation FA methyl esters and analyzed by the GC/MS technique. Positions of acyl substituents were determined by the sn-1 regioselective enzymatic cleavage of SQDGs using lipase type XI from *Rhizopus arrhizus*, followed by the isolation and analyses of products of enzymolysis. Exact positions of double-bonds in acyl substituents of **38**–**41** were established by gas chromatography/electron impact mass spectra of nicotinoyl derivatives prepared from the corresponding fatty alcohols, obtained as a result of LiAlH_4_ treatment of these glycolipids [67]. Structures of these and other microalgal SQDGs (**41**–**44**) are given in Table 2. This type of compounds, earlier isolated from other organisms, demonstrated anti-viral activity against HIV and antitumor properties [68].

The heterotropic dinoflagellates *Oxyrrhis marina* and *Gymnodinium dominus* are efficient producers of long-chain polyenic eicosapentaenoic (20:5n-3) and docosahexaenoic (22:6n-3) FAs. *O. marina* was cultured and used to isolate a new sulfoquinovosyl-containing glycolipid, the first SQDG (**41**) with highly polyunsaturated FA residue (ω-3 docosahexaenic acid), which was structurally elucidated by MS and NMR analysis, including 2D NMR experiments [69].

The sulfolipid mixture with **42** and **43** as main components was isolated by LH-20 chromatography from the methanol extract of the diatom *Thalassiosira weissflogii* with subsequent radial silica chromatography on Cromatotron and analyzed by NMR and LC/MS/MS technique [70,71,72]. Using these and accompanying glycolipids as model compounds, a novel immunomodulatory compound named β-SQDG18 (**44**) was synthesized and proved to activate human dendritic cells (hDCs) by TLR2/TLR4-independent mechanism. This immunoactive agent stimulated the maturation of dendritic cells (DCs) and triggered an immune response in vivo by the upregulation of MHC II, co-stimulatory proteins CD83, CD86, and pro-inflammatory cytokines L-12 and INF-γ. In the experimental melanoma model, vaccination of C57BL/6 mice with β-SQDG18-adjuvanted gelatin peptide (hgp10 peptide) induced a protective response with a reduction in tumor growth and an increase in survival of animals. These studies have opened a new class of adjuvants based on a single glycolipid molecule for application in immunology [71].

### 3.3. Deacylated Glycolipids

The biomass suspensions of the green microalga *Desmodesmus subspicatus* stimulated tomato (*Solanum lycopersicum*) seedlings via a foliar spray. A subfraction, obtained from the aqueous extract of this microalgae by chromatography on a Bio-Gel P-2 column, eluted with ultrapure water contains deacylated glycolipids 6-sulfo-α-D-quinovopyranosyl-(1→1)-glycerol (**45**), α-D-galactopyranosyl-(1→6)-β-D-galactopyranosyl-(1→1)-glycerol (**46**), and β-D-galactopyranosyl-(1→1)-glycerol (**47**) (Figure 6) along with the known zeatin. The subfraction induced increased hypocotyl lengths and volumes of seedlings, in comparison with standard treatment (water) and provided the same effect as a commercial product produced from the brown alga *Ascophyllum nodosum*. It is considered that, at least partly, these compounds are responsible for plant growth biostimulatory action and have a prospect to be used in the so-called organic agriculture [73]. 

## 4. Phosphoglycolipids

The microalga *Thalassiosira weissflogii* (CCMP 1336) was selected for the search for new immunomodulatory natural products, using screening of marine extracts with the use of human peripheral blood mononuclear cells (PBMC). In fact, extracts of this microalga induced IL-6 cytokine production in PBMC. A novel minor phosphogalactodiacylglycerol (PGDG-1, **48**), containing a phosphate group, galactopyranose and two diacylated glycerol residues, was isolated from the methanol extract of the cultured strain of *T. weissflogii* by bioassay-guided fractionation using LH-20 gel chromatography with subsequent HPLC. Its structure was determined by NMR and HR ESI MS in comparison with a model compound PGDG-2 (**49**), obtained by the 15-step synthesis from the commercial D-galactose. PGDG-1 exhibited immunostimulatory activity in human dendritic cells, while compound **49** showed Toll-like receptor-4 agonistic activity in human and murine dendritic cells and caused antigen-specific T-cell activation. The PGDG-1 was tested on monocyte-derived dendritic cells (moDCs) at the concentrations of 10–50 µg/mL and induced upregulation of surface markers at the lowest concentrations. When this substance was examined using moDCs, their maturation was stimulated. Increased expression of interleukins IL-6, IL-8, and IL-12p40 was observed beginning from the concentration of 5 µg/mL. The synthetic compound (**49**) showed a more remarkable effect on the upregulation of characteristic moDCs markers and the co-stimulatory molecules CD83 and CD86. A scheme of possible biogenesis of PGDGs from phosphatidyldiacylglycerol (**50**) and galactosyldiacylglycerol (**51**) by the reaction of transphosphatidylation was proposed and given below (Figure 7) [74]. 

A search platform was designed to identify novel low molecular weight natural products with anticancer immunotherapeutic activity. This platform includes an immunophenotypic assay using a growth factor–dependent immature DC line derived from a mouse spleen and direct antitumor property evaluation against nine tumor cell lines. After confirmation of both activities, the extracts that induced DC maturation and did not demonstrate nonspecific cytotoxicity were tested using DCs, generated in vitro from peripheral blood CD14+ monocytes [75]. To assess the selectivity of this screening methodology, extracts of the marine diatom *T. weissflogii* containing α-sulfoquinovosyl diacylglycerols (α-SQDGs) and atypical phosphoglycolipids (PGDGs) were subjected to further investigation. Some obtained results were discussed above, and others are in progress. 

## 5. Glycosides from Microalgae

### 5.1. Steryl Glycosides and Other Glycosylated Derivatives from Microalgae

Free sterols are important membrane constituents of eukaryotic organisms but are rarely present in prokaryotes, and well-studied. Sterol-conjugated containing carbohydrates are divided into two structural types: glucosides of sterols (**52**), predominantly of Δ^5^ series, and their acylated-by-fatty-acids derivatives (**53**) (Figure 8). In acylated derivatives, the acyl group occupies the C-6 position in a sugar moiety. These glucose-containing metabolites are frequently accompanied by other conjugated forms, such as steryl sulfates, and by diverse bioactive oxidized sterol derivatives [76]. Steryl glucosides modify biological membranes and provide resistance against temperature stress, regulate the host defense against pathogens and lipid metabolism, and support the development of different organisms. Cholesteryl acyl glucosides (**53**) stimulated lymphocytes in a CD1-dependent manner and are able to protect newborn animals against allergy [77].

Continuing our review [4], herein we report data concerning steryl glycosides that appeared in the literature after 2018. Recently, it was shown that steryl glycosides, recovered from biodiesel tanks after the use of terrestrial plant materials for fermentation, are excellent sources of valuable plant sterols [78]. Taking into account that the third generation of biodiesel production is based on the use of microalgae, it is of interest to examine whether precipitates forming at the fermentations of microalgal oils also contain steryl glycosides.

Recently, modern methods of structural biology and biotechnology were used to provide the production of such glycosides as rhamnolipids by microalgae. Rhamnolipids are metabolites of some bacteria that are used for bioremediation, enhancement of oil recovery and as biodegradable emulsifiers, foaming agents, detergents and cleaners. Such surface-active agents as rhamnosides of 3-hydroxyacylacids may contain one or several rhamnoside residues. The green microalga *Chlamydomonas reinhardtii*, approved by the US Food and Drug Administration as a food additive, was selected as a platform for the synthesis of 3-(hydroxydecanoyloxy)-decanoic acid HAA (**54**), a biosynthetic precursor of biosurfactant mono-rhamnolipid (**55**) (Figure 9). 

Chloroplast engineering was used to incorporate genes encoding RhIA acyltransferase that catalyzes the condensation of two 3-hydroxyacyl acids intermediates from the bacterium *Pseudomonas aeruginosa* into *C. reihardtii* to produce HAA lipid moiety of biosurfactants [79]. Nowadays, microalgae are more and more often studied as promising platforms for the biological synthesis of valuable metabolites not peculiar to these microorganisms themselves.

### 5.2. Glycosylated Long Chain Polyketide Derivatives 

Unique glycosylated polyketide derivatives have been found in several microalgal species. For example, amphidinins D and F (**56, 57**) belong to the structural type of the so-called amphidinins and are 4,5-secoanalogues of the previously known 12-membered macrolide amphidinolide Q, isolated from the culture of a dinoflagellate *Amphidinium* sp., which in turn was isolated from the marine acoel flatworm *Amphiscolops* sp. (Figure 10). The aglycones of these polyketides may be biosynthesized using unique biochemical machinery for a long chain extension by one carbon atom. These compounds contain α-D-ribofuranose residues in the carbohydrate moiety and exhibit moderate antimicrobial and antifungal properties, inhibiting the fungus *Trichophyton mentagrophytes* at concentrations of 16–32 μg/mL [80].

*Prymnesium parvum*, the microalgae belonging to the mixotrophic subphylum Haptophyta is one of the most known species causing significant economic losses in fishery. *P. parvum* produces glycosylated long-chain polyether toxins called prymnesins. Due to their hemolytic action, these ichthyotoxins cause massive fish kills in coastal and brackish waters. They were isolated, and the corresponding structures were determined a long time after the toxicity of *P. parvum* was first detected. In fact, studies on the toxic properties of this microalga were first reported in the Netherlands in 1920 [81] (cited from the recent review article [82]). Therefore, the studies on toxic *P. parvum* have been ongoing for 100 years. More than seventy years after this discovery, Japanese scientists determined the structures of the first purified toxins of this type, prymnesins-1 and -2 [83,84], as a result of extremely difficult isolation, purification, and careful NMR and MS studies of these toxins. The microalga was cultured, and about 400 L of the culture was used to isolate 10 mg of prymnesin-1 (**58**) (PRM-1) and 15 mg of prymnesin-2 (**59**) (PRM-2) from the corresponding extract (Figure 11). However, the poor solubility of the obtained toxins made difficult the application of NMR spectroscopy for structural studies of these compounds. To improve the solubility, prymnesins were converted into N-acetates by the reaction with Ac_2_O in an i-PrOH-water mixture (3:2). In addition, ^13^C-enriched toxins were also obtained by culturing this microalga in the medium containing Na_2_^13^CO_3_ to facilitate the use of ^13^C NMR spectroscopy. Peracetylation, hydrogenation and dehalogenation were applied to obtain the corresponding derivatives and to determine the number of free hydroxy groups, multiple double bonds, and other structure peculiarities of prymnesins. COSY, DGF-COSY, TOCSY and other NMR spectra revealed several partial structures. Their connection with each other in the total structure of prymnesin-2 was established by HMBC experiments. Performed analyses indicated the presence of conjugated double and triple bonds, chlorine and nitrogen atoms in the aglycon moiety of these molecules. Configurations of C14 and C76−C85 asymmetric centers remained unknown. Examination of the NMR data of N-acetylprymnesins as well as analysis of their carbohydrate portions with wide application of NMR spectra, comparison with literature data, and other approaches indicated that PRM-1 contains α-D-ribofuranosyl, α-L-arabinopyranosyl, and β-D-galactofuranosyl residues at C77, C78, and C82 positions, respectively. Another toxin, PRN-1, contains only one, but also rare monosaccharide, α-L-xylofuranose attached to C77.

Structure elucidation of polyether long-chain prymnesins-1 and -2 was an impressive achievement in marine bioorganic chemistry in the nineties. After partial synthesis, structures of prymnesins were refined and reassignment of the relative configurations in the E/F ring juncture was reported [85,86]. Later, closely related prymnesins began to classify as A and B types, respectively.

**Figure 11 marinedrugs-21-00427-f011:**
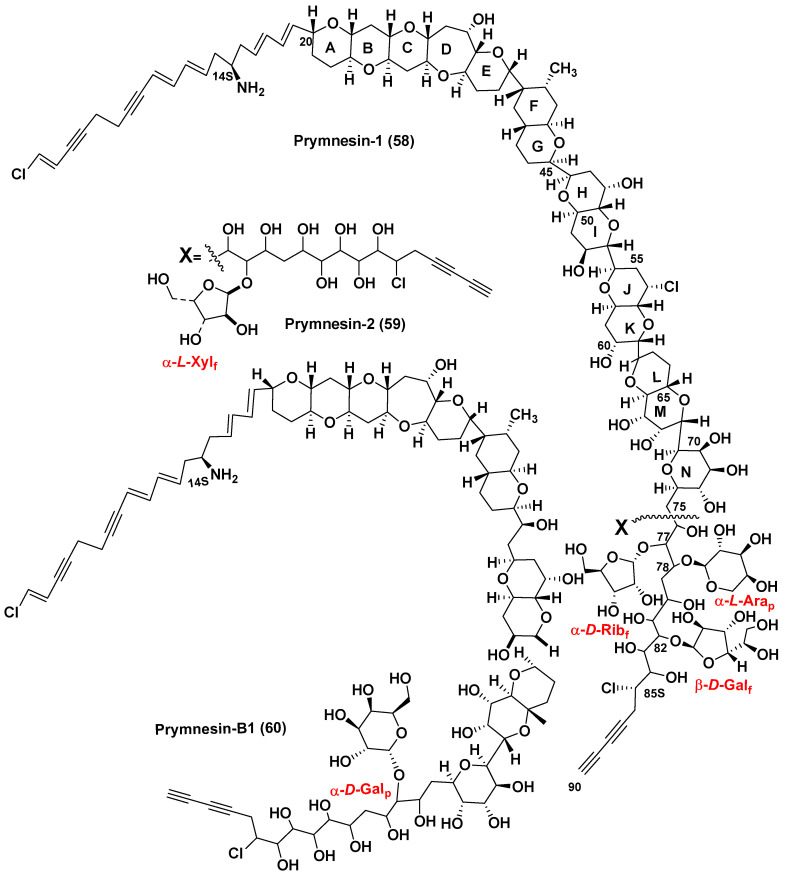
Prymnesins (**58**–**60**) [83,84,87].

Recently, Rasmussen et al. [87] used LC/HR MS to search for prymnesins in 10 strains of *P. parvum* collected worldwide. Authors found that only one of the studied strains produced the original prymnesins-1 and -2 (**58**, **59**), earlier isolated by the Japanese scientists. In addition, they reported the isolation and structure determination of prymnesin B1 (**60**) (Figure **11**). To isolate prymnesin B1, the authors used 90% ^13^C enriched initial materials and 2D- and 3D-NMR spectroscopy. Prymnesins of this type lack one 1,6-dioxadecalin unit and have a short acyclic C2 linkage in comparison with the original prymnesins. The nature of a sugar unit was determined using NMR and GC/MS approaches and acid-hydrolyzed **58**. The sugar in the hydrolysate of the ^13^C-enriched sample of the toxin was identified as D-galactopyranose by GC/MS of oxime-TMS derivatives and by the comparison with sugar standards followed by derivatization with N-methyl-bis-trifluoroacetamide and analysis of the obtained trifluoroacetyl derivative by the chiral-phase GC/MS as well as by long-range HSQC experiments. The precise position of this sugar at C71 in the core structure and the α-stereochemistry of the glycoside bond were confirmed by the deshielded ^13^C chemical shift (δ^13^C = 90.3 ppm) of the anomeric carbon, ^3^J_H-1′,H-2′_ = 4 Hz and NOE correlation between H-1′ and H71. Like prymnesin-2, this novel B-type prymnesin was extremely toxic against an RTgill-W1 cell line by inhibiting the growth of these cells at low nanomolar concentrations. 

Further investigation of 26 strains of *P. parvum,* collected worldwide, for the presence of prymnesins using the LC/HR MS approach established that only four examined strains contained **58** and/or **59**. Other prymnesin analogs differed in their backbones, chlorination, and glycosylation patterns were tentatively detected by the LC/MS/MS, and C-type prymnesins were found in five studied strains. C-type prymnesins, a new type of these toxins, were proved to contain a C83 backbone, but their full structural analysis was not completed so far. Thus, it was indicated that the evolution of prymnesins has led to a diverse family of fish-killing glycosylated toxins, having a significant ecological influence on inhabitants of coastal brackish waters [88].

Prymnesins demonstrate potent hemolytic, cytotoxic, neurotoxic, and ichthyotoxic activities at nanomolar concentrations. Structurally, these ichthyotoxins, due to their biphylic structures, are detergent-like molecules as they have both polar and non-polar moieties like some known steroid and/or triterpene glycosides. They damage the gill-breathing marine organisms predominantly and induce ion leakage and cell lysis, followed by the death of fish and mollusks. Environmental factors such as salinity, pH, ion availability, and growth phase influence the prymnesin accumulation and toxicity of *P. parvum* strains [89].

During red tides and later, prymnesins affect different surrounding hydrobionts, including competitors and grazers. Prymnesins paralyze or kill a plethora of marine inhabitants [90]. Using its toxins, *P. parvum* can kill and eat the predatory heterotrophic dinoflagellates *Oxyrrhis marina*. As a result of the action of these toxins, cells of *O. marina* lose normal shape, and are leased, while particulate materials originating from destroyed *O. marina* are ingested by *P. parvum* [91].

Details of biosynthesis of these glycosylated toxins remain not established, although a cDNA library was constructed from late log-phase cultures of *P. parvum* with 3415 unique tentative unigenes [TUGs] found. Many of these TUGs, including 12 of the 50 most commonly encountered transcripts, encoded novel proteins, probably involved in the synthesis and secretion of prymnesins [92]. In a very recent investigation, the transcriptomes of nine strains of *P. parvum* strains were analyzed, and numerous genes of polyketide synthases (PKS I) were found. Eight consensus transcripts were present in all these nine strains examined. The detailed analysis of PKSs in *P. parvum* is the next step toward a better understanding of the biosynthesis of prymnesins [93].

The structure diversity of prymnesins suggests that the evolution of the polyketide biosynthetic machinery in microalgae has led to a variety of prymnesin backbone structures. As suggested by previous investigations on the biosynthesis of algal polyketide in dinoflagellates, the formation of the backbone polyketide chains in microalgal toxins is realized by not only a simple extension of the growing chain with two carbons of intact acetate units but also by condensation with the cleaved acetate and glycolate units [94].

Prymnesins are biosynthesized more effectively in response to physiological stress conditions such as low salinity or high irradiation. During stress conditions, the expression of selected PKS genes is increased. This confirmed the adaptive role of prymnesins and the role played by PKSs in the biosynthesis of prymnesins and cell adaptation [95].

## 6. Viral Regulation of Microalgal Blooms and Glycoconjugates

Recently, much attention has been attracted to issues related to the regulation of harmful microalgae blooms as a result of viral infections. The abundance of viruses as the most common entities in the world’s oceans (ten billion viruses per liter of seawater) exceeds that of bacteria. Thirty years ago, Japanese scientists reported that viral particles presented in microalgae 3 days before the end of the red tide in the Northern part of Hiroshima Bay. Later, the strain-specific virus (HaV, *Heterosigma akashiwo* virus), clone GSNOU-30 was isolated and suggested that the virus plays the role of selector, increasing the genetic diversity of host microalgae [96,97]. 

A boom in the studies on marine viruses, which regulate the dynamics of microalgal blooms, has been continuing for more than two decades. Viruses terminate blooms by destruction of microalgae via mechanisms resembling the programmed cell death in metazoans with DNA fragmentation and activation of cysteine aspartate-proteases (caspases). Particular attention was given to the regulation of blooms caused by the microalgae, participating in global geochemical cycles of carbon, calcium, and sulfur, as well as by harmful microalgae [98]. For example, the viral termination was established in relation to the bloom events caused by coccolithophore *Emiliania huxleyi* and the virus was isolated [99,100]. This alga is a producer of one-third of the total marine CaCO_3_ production, and in addition, it also releases dimethyl sulfide into the atmosphere, which enhances cloud formation and influences climate. Viral infections determine the cell fate in coccolithophore populations as well as the host-virus dynamics [101].

The *Emiliania* giant DNA viruses (EhVs) are surrounded by lipid envelopes composed of glycosphingolipids (GSLs) with a smaller proportion of polar glyceroglycolipids such as MGDG, DGDG, and SQDG. The alga is enriched in glycolipids, which make up 65% of their lipidome. Since viral infection involves membrane fusion between the virus and host, it was postulated that specific membrane lipids may facilitate the attachment of the virus. Sphingolipids were proved to be a crucial factor for a successful viral infection, followed by the sudden crashes of *E. huxleyi* blooms. Actually, the *E. huxleyi* populations, which have the greatest content of glycerosphingolipids (GSLs), showed the highest rate of virus-induced mortality. A potential role of GSLs is to regulate the sensitivity and resistance in the dynamics of the *E. huxleyi*-EhV system [102]. The virus genome penetrates into the host and changes the biosynthesis of its GSLs. Viral serine palmitoyltransferase uses as a substrate C15-CoA instead of C16-CoA. This leads to the formation of a shortened sphinganine base (C17 instead of C18) and to gluconjugates like (**61**) instead of host glycosphingolipids, such as **62** (Figure 12).

Generally, the bloom termination is a result of the horizontal transfer of genes, encoded sphingolipid biosynthesis pathway, between the virus and eukaryotic host. Viral metabolites cause the metabolic shift in sphingolipid biosynthesis of the host microalga that leads to the death of microalgae [102,103]. The genome of *E. hexleyi* has a variety of unexpected genes involved in the biosynthesis of sphingolipids and glycosphingolipids [104]. Fulton et al. [105] identified intact glycosphingolipids with tentative structure **63** (Figure 12) in this microalga. They noted that *E. hexleyi* populations with the greatest content of such glycosphingolipids had the highest rate of mortality due to viral infection [105]. 

Recently, another large double-stranded DNA virus (PpDNAV) was isolated from the site of a harmful bloom event of *Prymnesium parvum* in Norfolk, England. The virus was found in 5 out of 15 studied tested strains of the microalga and lysed >95% of sensitive strains by 120 h post-infection. Using phylogenetic clustering, it was shown the structure, and probably the action of this virus can be phylogenetically similar to those of other algal viruses belonging to the Megaviridae family [106].

## 7. Carbohydrate-Containing Metabolites of Microalgae and Deep-Sea Life

A huge amount of microalgal galactolipids, the most common secondary metabolites in plants, enter the seawater and the bottom of oceans and are easily absorbed by bacteria and invertebrates that feed on water-soluble and suspended organic matter and detritus and live in deep-sea environments. In fact, marine phytoplankton excretes photosynthate, mainly consisting of carbohydrate-containing metabolites, into seawater in the form of dissolved organic matter. Radioactivity of 7–50% was found within 7–24 h in dissolved organic matter in the water after the addition of ^14^C-bicarbonate to the samples of lake or sea waters. Production of extracellular ^14^C-labelled organic matter was found in different types of phytoplankton [107,108]. Dead organic matter (detritus) is accumulating in deep-water continuations of river flows, in underwater intermountain areas and on the slopes of seamounts. Detritus plays an important role as a resource and a habitat for many deep-sea species and provides stability to biosystems by regulating their trophic structures and biodiversity [109].

## 8. Some Perspectives

Microalgae are excellent model organisms to develop different techniques of modern omics glycobiotechnology [110], bioengineering and structural biology. This opens new frontiers in improving the corresponding strains of microalgae for obtaining valuable products and discovering new directions of biosynthesis and metabolism in these lower plants. 

## 9. Conclusive Remarks

Microalgae have attracted a lot of attention due to the possibility of being used for the industrial production of biofuels and as a source of the components of nutritionally healthy food and new drugs. Carbohydrate-containing natural products from microalgae have become one of the focuses of research that leads to obtaining new knowledge about the structures, biological activity, functions and taxonomic distribution of glycolipids, various glycosides, glycosylated toxins and pollutants such as ribosylated arsenic compounds. Relevant studies are focused on such phenomena as the regulation of microalgae blooms and poisoning of humans, fish and invertebrates with phycotoxins. 

A variety of monosaccharide residues in microalgal carbohydrate-containing metabolites considered in the review is quite large and includes a number of rare sugars and sugars connected to other parts of molecules not by common β–, but α-glycoside bonds (Table 3). Particularly, diverse and unique carbohydrate moieties were found in toxins. 

The screening of new carbohydrate-containing metabolites of marine microorganisms has so far affected only a small part of the currently known species due to problems with obtaining sufficient biomass for research and difficulties in cultivating some microalgae. As a rule, the procedures for isolation and separation of complex mixtures of polar metabolites take a lot of time. 

Summarizing the data on the biological activities of the metabolites considered in this review, it may be concluded that glycosylated toxins and some immunoactive compounds are of the greatest interest as biologically active agents. In our opinion, phoshphoglycolipids, which cause antigen-specific activation of T-cells, and sulfoglycolipid adjuvants, studied by the Italian group from Napoli, are the most promising among immunomodulators.

## Figures and Tables

**Figure 1 marinedrugs-21-00427-f001:**
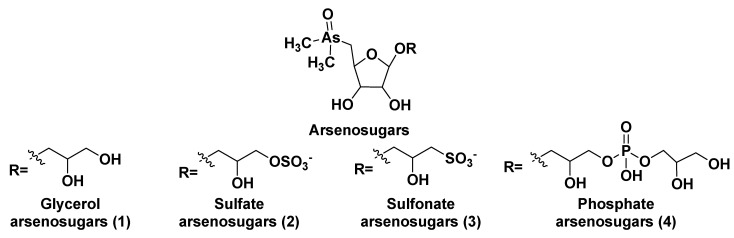
Main types of arsenosugar derivatives (**1**–**4**).

**Figure 3 marinedrugs-21-00427-f003:**
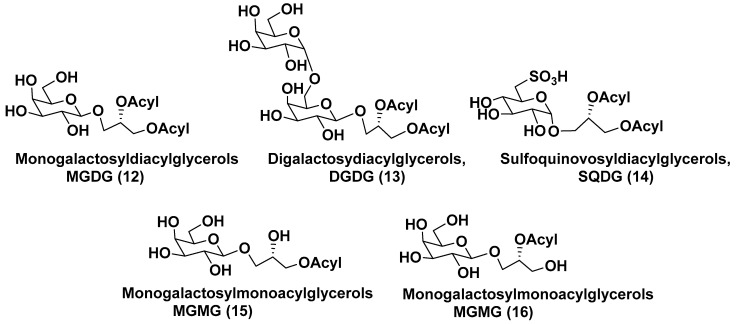
Main structural types of galactolipids (**12**–**16**).

**Figure 4 marinedrugs-21-00427-f004:**
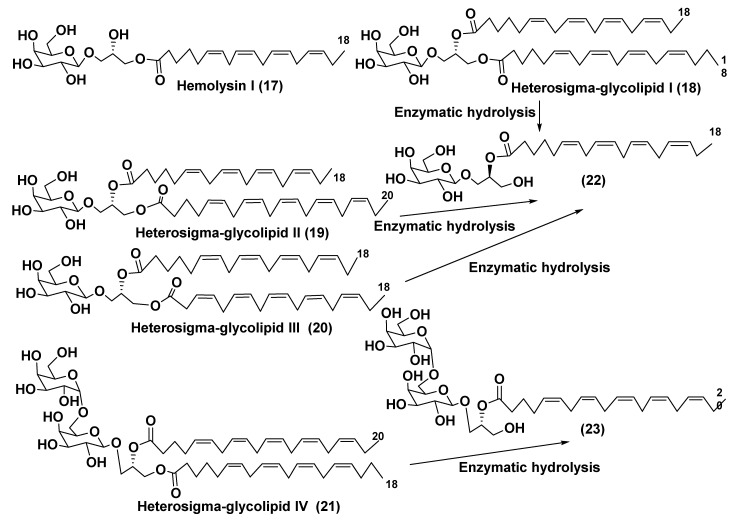
Structures and enzymatic conversions of galactolipids (**17**–**23**) [28,29].

**Figure 5 marinedrugs-21-00427-f005:**
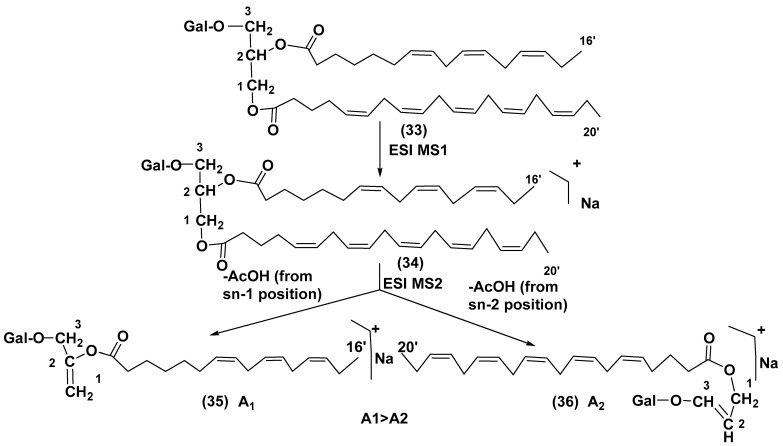
Scheme of fragmentation of galactolipids at ESI MS of 33 [47].

**Figure 6 marinedrugs-21-00427-f006:**
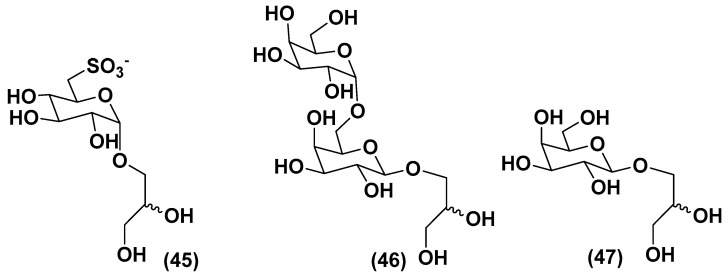
Deacylated glycolipids from the green microalga *Desmodesmus subspicatus* (**45**–**47**) [73].

**Figure 7 marinedrugs-21-00427-f007:**
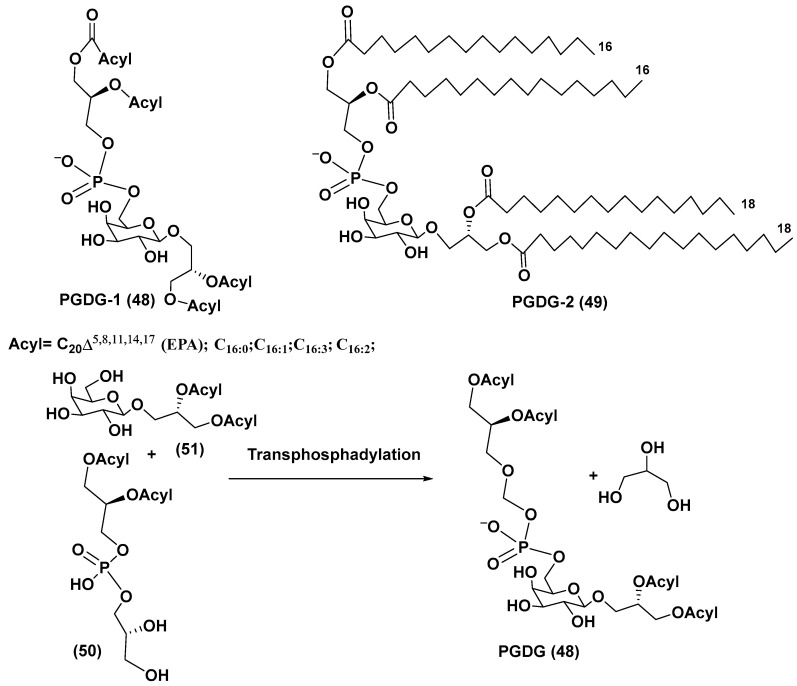
Phosphogalactodiacyl glycerols and biogenesis of these metabolites (**48**–**51**) [74,75].

**Figure 8 marinedrugs-21-00427-f008:**
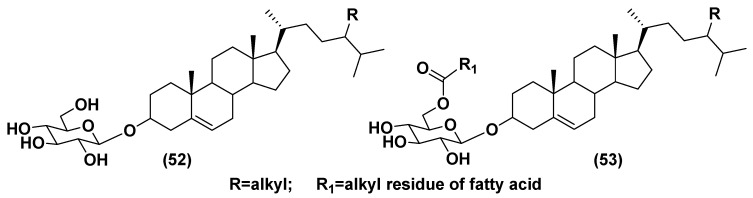
General formulae of steryl glucosides (**52**–**53**) [76,77].

**Figure 9 marinedrugs-21-00427-f009:**
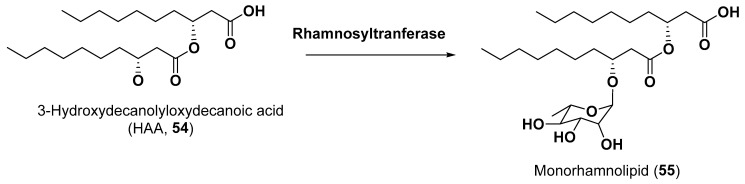
Biogenesis of monorhamnolids (**54**–**55**) [79].

**Figure 10 marinedrugs-21-00427-f010:**
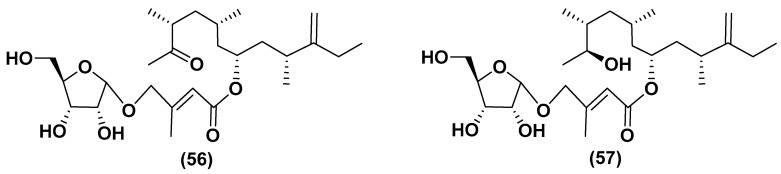
Amphidinins D and F (**56**,**57**) [80].

**Figure 12 marinedrugs-21-00427-f012:**
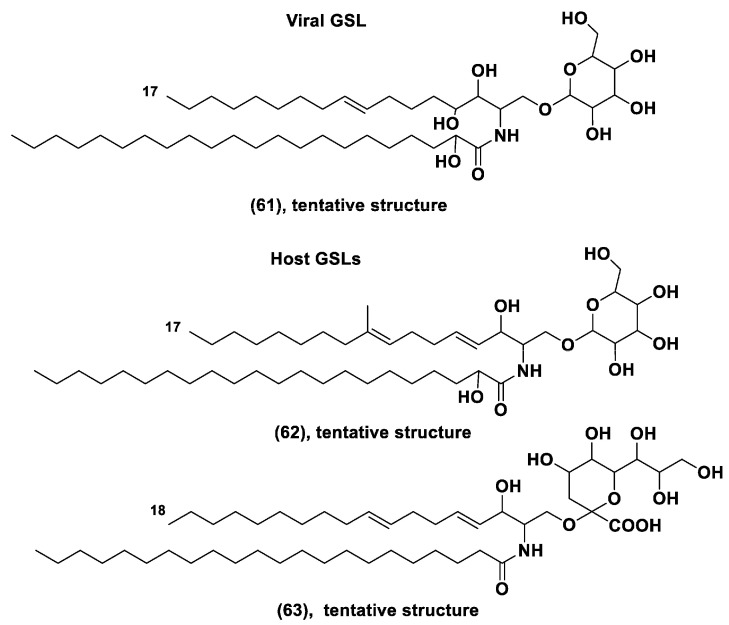
Unusual glycosphingolipids from *E. hexley* virus and *E. huxleyi* (**61**–**63**) [102,103,104].

**Table 1 marinedrugs-21-00427-t001:** Structures and activities of some galactolipids from microalgae.

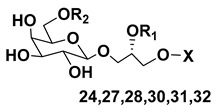		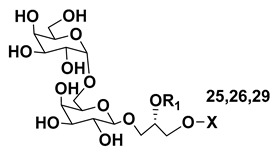		
Compounds, Source, [Ref.]	R_2_=	R_1_=	X=	Activity
**24**, *Heterocapsa circularisquama* [30]	H	H	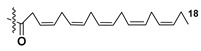	Cytolytic activity towards heart and gill cells of oyster at a concentration of 6.2 mg/mL
**25**, *Heterocapsa circularisquama* [30]	H	H	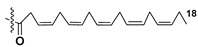	Cytolytic activity towards heart and gill cells of oyster at a concentration of 6.2 mg/mL
**26**, *Heterocapsa circularisquama* [30]	H	H	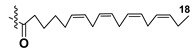	Cytolytic activity towards heart and gill cells of oyster at a concentration of 6.2 μg/mL
**27**, *Hymenomnas* sp., [31]	H	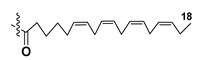	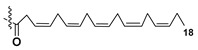	Inhibition of Na^+^,K^+^ -ATP-ase with IC_50_ 2 × 10^−5^ M
**28**, *Hymenomonas* sp., [31]	H	-	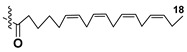	Inhibition of Na^+^,K^+^ -ATP-ase with IC_50_ 2 × 10^−5^ M
**29**, *Hymnomonas* sp., *Aphidinum carterae* [31,33]	H	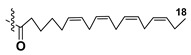	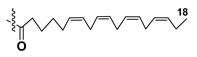	Inhibition of Na^+^,K^+^ -ATP-ase with IC_50_ 2 × 10^−5^ M
**30**, *Amphidinium* sp., [32]	C_18:0_	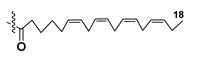	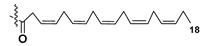	
**31**, *Scrippsiella trochoidea*, [34]	H	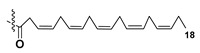	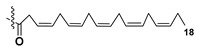	Strong inhibitory action against Ca^+2^ ion-influx in rabbit platelet cells
**32**, *Karenia mikimotoi,* [35]	H	C_14:0_	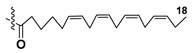	Significant reduce of expression of CD124 in RAW 264.7 macrophages activated by LPS

**Table 2 marinedrugs-21-00427-t002:** Structures and activities of sulfoglycolipids **37**–**44**.

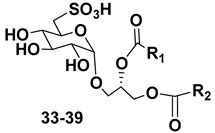
Compounds, Source [Ref.]	R_1_	R_2_	Activity
**37**,*Heterosigma carterae,* [67]	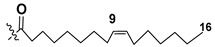	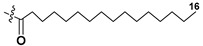	
**38***Heterosigma carterae,* [67]	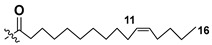	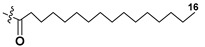	
**39***Heterosigma carterae,* [67]	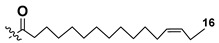	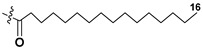	
**40**,*Heterosigma carterae,* [67]	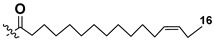	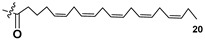	
**41**,*Oxyrrhis marina,* [69]	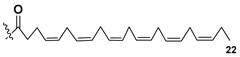	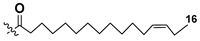	Significant NO inhibitory effect in LPS-activated RAW264.7 cells without affecting cell viability
**42**,*Thalassiosira weisflogii*, [70]	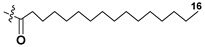	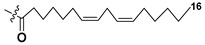	Induction of IL-12 and HLDA-DR overexpression at a concentration of 10 ng/ML
**43***Thalassiosira weisflogii*,[70]	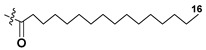	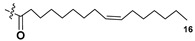
β-SQDG18 **44**,Sulfavant A(synthetic), [70]	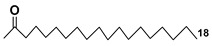	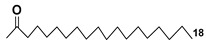	Induction of maturation of dendritic cells with the upregulation of expression of MHC II, co-stimulatory proteins (CD83, CD86), and cytokines IL-12 and INF-γ.

**Table 3 marinedrugs-21-00427-t003:** Some monosaccharides are found in glycoconjugates of microalgae.

Group of Metabolites	Sugars	Ref.
Arsenosugars	5–dimethylarsenoyl-β-D-Rib_f_	[10,11,12,13]
Arsenolipids	5–dimethylarsenoyl-β-D-Rib_f_, 5–dimethylarsenoyl-2-O-methyl-β-D-Rib_f_	[15,17,18,19]
Galactolipids	α– and β–D-Gal_p_	[29,30,31,32,33,34,35]
Sulfoquinovosyl-containing glycolipids	6-sulfo-α-D-Gui_p_	[37,38,39,40,41,42,43,44]
Phosphoglycolipids	β–D-Gal_p_	[74]
Steryl glycosides	D-Glc_p_, 6-O-acyl-D-Glc_p_	[4]
Amphidinins	α–D-Rib_f_	[80]
Prymnesins	α–L-Xyl_f_, α–D-Rib_f_, α–D-Gal_p_, β–D-Gal_f_, α–L-Ara_p_	[83,84,87]

## Data Availability

The data presented in this study were available on request from the corresponding authors and from open databases.

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
