# Peer review of "Carbohydrate-Containing Low Molecular Weight Metabolites of Microalgae"

_marinedrugs, 2023, doi:10.3390/md21080427_

Round 1

Reviewer 1 Report

This review focused on the isolation, structure determination, functions and biological activities of carbohydrate-containing natural products, including glycosylated arsenicals, glycolipids, phosphoglycolipids, glycoside derivatives and other groups of glycoconjugates from microalgae. This review is well organized, well concluded the glycoconjugate areas of microalgae. It is comprehensive and relevance to the field of carbohydrate containing-natural product. 

However, the format of the references should be carefully checked. For example, line 835, comma follow the Biomolecules, line 838, Blood Rev.  

I recommend it accept after minor revision. 

Overall, the language is easy to read.  Typos should be checked.

Line 802, bioingeeniring should be bioengineering. 

Author Response

Authors’ answers to Reviewer 1 comments 

REVIEWER 1.

Dear Reviewer,

Thank you very much for your comments.

Please find author’s response to your comments below. 

  1. The format of the references should be carefully checked. For example, line 835, comma follow the Biomolecules, line 838, Blood Rev

The reference list was carefully checked.

  1. Line 802, bioingeeniring should be bioengineering. 

It was corrected.

We are grateful to reviewer for useful comments.

Kind regards,

Inna Stonik

Corresponding author
A.V. Zhirmunsky National Scientific Center of Marine Biology,

Far Eastern Branch, Russian Academy of Sciences.
Pal'chevskogo Str., 17,
Vladivostok 690041, Russia

Reviewer 2 Report

The manuscript entitled “Carbohydrate-containing Low Molecular Weight Metabolites 1 of Microalgae” is a Review describing mainly the structure of glycoconjugates of low molecular mass from microalgae.

The manuscript presents many problems. Firstly, it is hard to read since only a description without using a table has been chosen. The result is a very boring reading which obligates the reader to go up and down for finding a sense in the manuscript.

Secondly, there are many mistakes in the chemical formula, see for example Figure 15 (Page 21, line 764), where the authors inserted the formula (55), describing it as a “sialo-GSLs”. The sugar is not a sialic acid, which in turn is not “2-keto-3-deoxy-D-glycero-D-galacto-nononic acid” (Line 753, page 21). Or for example, on Page 14, line 521, compound (37), which is not a quinovose configured sugar. Many others of this type of mistake are found everywhere in the manuscript.

Thirdly, a paragraph containing the structure/activity relationships of these compounds should be inserted.

Finally, and not less importantly, the English language should be revised by a mother tongue.

For all these reasons, I think that the manuscript should be rewritten, and therefore my suggestion is a rejection.

The English language should be revised by a mother tongue. In many sentences, authors omit prepositions, articles and verbs.

Author Response

Authors’ answers to Reviewer 2 comments 

REVIEWER 2.

Dear Reviewer,

Thank you very much for your comments. The manuscript was carefully revised and partly rewritten according your recommendation.  We send the pdf file of the revised manuscript. In addition, we also send the revised version of the manuscript as *doc file with our corrections highlighted by green.

Please find author’s response to your comments below. 

  1. It is hard to read since only a description without using a table has been chosen. The result is a very boring reading which obligates the reader to go up and down for finding a sense in the manuscript.

We inserted three new tables, where we summarized data concerning structures and activities of galactolipids from microalgae (Table 1), the corresponding data concerning sulfoglocolipids (Table 2), and some monosaccharides found in glycoconjugates (Table 3).

  1. There are many mistakes in the chemical formula, see for example Figure 15 (Page 21, line 764), where the authors inserted the formula (55), describing it as a “sialo-GSLs”. The sugar is not a sialic acid, which in turn is not “2-keto-3-deoxy-D-glycero-D-galacto-nononic acid” (Line 753, page 21). Or for example, on Page 14, line 521, compound (37), which is not a quinovose configured sugar. Many others of this type of mistake are found everywhere in the manuscript.

The formula (55) was corrected, the term “sialo-GSLs” as well as “2-keto-3-deoxy-D-glycero-D-galacto-nononic acid” were deleted. The formula (45) (the former 37) in new Figure 6 was corrected too. We also checked other formulae and the text, and corrected several mistakes.

  1. A paragraph containing the structure/activity relationships of these compounds should be inserted.

The information concerning structure/activity relationships are inserted in new Tables 1-2, and concluded in the paragraph beginning from “Summarizing the data on the biological activities of the metabolites…” (the last paragraph of Conclusive remarks.

  1. Finally, and not less importantly, the English language should be revised by a mother tongue.

English was revised by native speaker.

We are grateful to reviewer for useful comments.

Kind regards,

Inna Stonik

Corresponding author
A.V. Zhirmunsky National Scientific Center of Marine Biology,

Far Eastern Branch, Russian Academy of Sciences.
Pal'chevskogo Str., 17,
Vladivostok 690041, Russia

Reviewer 3 Report

In this manuscript “Carbohydrate-containing Low Molecular Weight Metabolites of Microalgae”. This review focuses on the isolation, structure determination, properties, and approaches to discovering new bioactive metabolites. It analyzes the existing literature on the structures, functions, and biological activities of various glycoconjugate groups, including ribosylated arsenicals, galactosylated and sulfoquinovosylated lipids, phosphoglycolipids, glycoside derivatives of toxins, and others. The review highlights the discovery of a wide range of new carbohydrate-containing metabolites and their important biological roles. Additionally, these metabolites exhibit potent immunomodulatory activity, which represents an interesting recent achievement. The structures, properties, and biological roles of glycoconjugates from microalgae continue to attract significant attention and research interest year after year. There are some problems in the manuscript, which could be revised further. The comments and problems are as follows: 

1. The source of the chemical structure diagram of the substance should be labeled with a reference or source.

2. A brief description of the synthesis of the compound can be added to the figure caption.

3. The compounds are labeled with Arabic numerals in the text, and a simple summary table can be included at the end of the text to facilitate understanding.

4. The compound figures in the text can be summarized with their isolation, structure determination, properties contents in tables or the layout can be improved to make them easier to understand.

5. The keywords could be more concise.

Author Response

Authors’ answers to Reviewer 3 comments 

REVIEWER 3.

Dear Reviewer,

Thank you very much for your comments.

Please find author’s response to your comments below.

  1. The source of the chemical structure diagram of the substance should be labeled with a reference or source.

Sources of the chemical structures and the corresponding references were inserted in structure diagrams, where it was possible.

  1. A brief description of the synthesis of the compound can be added to the figure caption.

As a rule, the described compounds were not synthesized, but they were isolated from microalgae. 

  1. The compounds are labeled with Arabic numerals in the text, and a simple summary table can be included at the end of the text to facilitate understanding.

Three new summary tables were inserted in the text (Tables 1-3).

  1. The compound figures in the text can be summarized with their isolation, structure determination, properties contents in tables or the layout can be improved to make them easier to understand.

As for instances, when the full structures were determined, the corresponding formulae were given in the text as well as in new Tables 1-3.

  1. The keywords could be more concise.

The number of keywords was shortened.

We are grateful to reviewer for useful comments.

Kind regards,

Inna Stonik

Corresponding author
A.V. Zhirmunsky National Scientific Center of Marine Biology,

Far Eastern Branch, Russian Academy of Sciences.
Pal'chevskogo Str., 17,
Vladivostok 690041, Russia

Round 2

Reviewer 2 Report

The manuscript has been ameliorated. Accepting is recommended.